# Ubiquitin and Ubiquitin-Like Proteins and Domains in Ribosome Production and Function: Chance or Necessity?

**DOI:** 10.3390/ijms22094359

**Published:** 2021-04-22

**Authors:** Sara Martín-Villanueva, Gabriel Gutiérrez, Dieter Kressler, Jesús de la Cruz

**Affiliations:** 1Instituto de Biomedicina de Sevilla, Hospital Universitario Virgen del Rocío/CSIC/Universidad de Sevilla, 41009 Seville, Spain; saramarvi@msn.com; 2Departamento de Genética, Universidad de Sevilla, 41013 Seville, Spain; ggpozo@us.es; 3Unit of Biochemistry, Department of Biology, University of Fribourg, CH-1700 Fribourg, Switzerland

**Keywords:** protein folding, ribosomal protein, ribosome biogenesis, SUMO, ubiquitin, ubiquitin-like domain

## Abstract

Ubiquitin is a small protein that is highly conserved throughout eukaryotes. It operates as a reversible post-translational modifier through a process known as ubiquitination, which involves the addition of one or several ubiquitin moieties to a substrate protein. These modifications mark proteins for proteasome-dependent degradation or alter their localization or activity in a variety of cellular processes. In most eukaryotes, ubiquitin is generated by the proteolytic cleavage of precursor proteins in which it is fused either to itself, constituting a polyubiquitin precursor, or as a single N-terminal moiety to ribosomal proteins, which are practically invariably eL40 and eS31. Herein, we summarize the contribution of the ubiquitin moiety within precursors of ribosomal proteins to ribosome biogenesis and function and discuss the biological relevance of having maintained the explicit fusion to eL40 and eS31 during evolution. There are other ubiquitin-like proteins, which also work as post-translational modifiers, among them the small ubiquitin-like modifier (SUMO). Both ubiquitin and SUMO are able to modify ribosome assembly factors and ribosomal proteins to regulate ribosome biogenesis and function. Strikingly, ubiquitin-like domains are also found within two ribosome assembly factors; hence, the functional role of these proteins will also be highlighted.

## 1. Introduction

Ubiquitin (Ub) is a highly conserved small protein of about 76 amino acids that is present in practically all eukaryotic cells. During evolution, its sequence, and even more so, its fold has been extremely conserved [1]. Ub-like proteins have also been identified in prokaryotes, providing clues on the origin of this ubiquitous protein [2]. The structure of Ub revealed a unique fold formed by a β-sheet with five antiparallel β-strands and a single helical segment, which is shared by other Ub-like proteins and domains, known as the β-grasp fold (Figure 1) [1,3].

Ub functions as a reversible post-translational modifier of proteins to regulate many different cellular processes. Since its discovery, a role of Ub in proteasome-dependent protein degradation has been emphasized (e.g., [5,6]), but beyond this function, Ub participates in many other cellular processes, such as DNA repair, chromatin dynamics, cell cycle regulation, membrane and protein trafficking, endocytosis, autophagy, and transcriptional and translational control (reviewed in [7,8,9]). Accordingly, Ub is a very abundant cellular protein that is used to modify a large number of different proteins in yeast (>1000) and human (>9000) cells [10,11]. The enzymatic conjugation of Ub to other cellular proteins is referred to as ubiquitination or ubiquitylation and requires specific Ub ligases, while, conversely, removal of Ub from modified targets is called deubiquitination and involves the activity of specific proteases; these aspects will only be briefly discussed here as they are the subject of more specialized review articles (see [5,7,12,13,14,15]). As a result of ubiquitination, the Ub moiety is attached via an isopeptide linkage to its substrate protein, involving the carboxyl group of Ub’s terminal glycine residue (G76) and an epsilon-amino group of a lysine residue in its substrate or in another Ub molecule (K6, K11, K27, K29, K33, K48, and K63) [16]. Occasionally, the carboxyl group of G76 can be covalently linked to the N-terminal methionine residue (M1) of another Ub molecule and, less frequently, to non-lysine residues within substrate proteins [16]. Thus, ubiquitination comes in different flavors: substrate proteins can be modified at one (mono-Ub) or multiple (multimono-Ub) site(s) by a single Ub or several Ubs, forming structurally distinct homo- or heterotypic linear, or even branched, chains; moreover, the attached Ub chains can also be modified by acetylation or phosphorylation, leading to a further expansion of the repertoire of functional consequences of ubiquitination [17,18]. The best understood role of Ub is probably the targeting of substrate proteins carrying single K48-linked polyUb chains to the proteasome for degradation [6]. However, some types of ubiquitination, especially the ones containing mixed or branched heterotypic Ubs, still await the assignment of a specific function [16,17]. In any case, it is clear that the different modes of Ub conjugation offer new possibilities and fates to the substrates, including the alteration of their inter- and intramolecular interactions, which could affect their localization, complex formation or dissociation, or activity [17], a concept that it is not limited to Ub but can also be applied to Ub-like modifiers (see below).

The Ub fold is widely distributed across eukaryotes and is even present in prokaryotes [1,14]. Basically, proteins containing the Ub fold can be classified, based on their ability to be covalently attached to substrate proteins or other molecules, into two categories, comprising either Ub-like or Ub-related modifiers (class I) or proteins harboring as part of their larger structures a Ub-like domain (class II) and, therefore, also being referred to as Ub-domain proteins [19,20].

(i) Proteins belonging to the first category share with Ub the presence of one to two glycine residues at their C-terminal end and internal lysine residues. As Ub (see later), these proteins are normally translated as immature precursors that must be processed by a specific protease to generate their mature forms, bearing an active carboxyl group on the exposed terminal glycine, which can be attached to a protein, normally via a lysine residue, or, exceptionally, a phospholipid substrate [21]. However, there are Ub-like modifiers that are not synthetized as precursors (e.g., Urm1 proteins). Most Ub-like conjugating enzymes and pathways resemble those involved in ubiquitination [22]. Prominent members among a larger list of Ub-like modifiers are SUMO (yeast Smt3), ISG15, NEDD8 (yeast Rub1), and ATG8 family proteins (yeast Atg8) [21,23]. Similarly, as observed upon ubiquitination, the covalent modification of substrates with Ub-like proteins enables functional regulation by altering their inter- or intramolecular interactions, thereby transforming and modulating their properties, including their structure, activities, or localization [15]. Two relevant features that will be discussed herein are the expression and stability of substrates and fusion proteins, natural or artificial, which can both be enhanced by attached Ub or Ub-like modifiers (e.g., [24,25,26]).

(ii) The second category of Ub-related proteins comprises proteins containing a Ub-like domain as an integral part of a longer protein from which it is not dislodged by a protease. As we will discuss later, the Ub-like domain of these proteins provides a specific important function, such as, for instance, a binding site for another protein [15]. 

In this review, we explore the role of Ub, Ub-like modifiers, and proteins containing Ub-like domains in ribosome biogenesis and function, with a particular emphasis on those findings reported in the yeast model organism *Saccharomyces cerevisiae*. 

## 2. The Ubiquitin Genes in Eukaryotes

Practically all Ub-like modifiers, as well as Ub itself, are synthesized by proteolytic maturation from precursor proteins. This maturation is required to release and activate the C-terminal glycine residue, which is essential for the conjugation onto its targets via a cascade of three different enzymes [22]. 

To study how Ub genes are arranged along the eukaryotic kingdom, we used the protein family database PFAM [27]. For this, we searched for the well-defined Ub domain (PF00240 entry), as eukaryotes that are evolutionarily far apart show a high degree of Ub conservation (e.g., human and yeast Ub proteins share 96% identity). However, and as expected due to the high structural homology among Ub and Ub-like proteins, this entry, in addition to the bona fide Ub domain, also contains a large number of diverse Ub-like and incomplete Ub domains. Thus, to properly perform this analysis, we first downloaded all 38,884 Ub domains comprised within the PF00240 entry of PFAM and compared them by BLAST [28] to the yeast Ub domain, which corresponds to the Ub moiety present in any of the four *UBI* genes present in yeast (see below). Appendix A shows the identity distribution found for this comparison. Two main groups of identity were found; for further analyses, we focused on the group most similar to the yeast Ub domain, thereby discarding all proteins belonging to the PF00240 entry whose Ub domain had less than 80% identity in at least 80% of its length with the yeast Ub. This procedure yielded a total of 5349 proteins (Appendix A). Then, using Perl scripts developed by us, we cross-referenced this list of proteins with the collection of protein families of PFAM, which are organized as architectures and, thus, composed of one or more domains in diverse combinations. In such a way, we evaluated the combination of the Ub domain with other domains, taking additionally into account the taxonomic information for each studied protein. As a result, we found that the Ub domain combines with 114 domains in 147 different architectures (see Appendix A). Amongst these, the 10 most common ones comprise about 92% of all Ub-containing proteins in three distinct architectural arrangements (Figure 2): 

(i) First, there are genes consisting of only a single copy of the Ub coding sequence; thus, the translation of the corresponding mRNA generates a monoUb precursor protein (11% of all architectures). Although it has been considered in the literature that this arrangement has only exceptionally been found in a small group of eukaryotes, such as two primitive single-celled intestinal parasites, *Giardia lamblia* and *Entamoeba histolytica* (Appendix A) [29,30], the true situation seems to be quite different as our analysis reveals that a variety of different animals, fungi, and plants also possess genes encoding a monomeric Ub domain (Figure 3). 

(ii) The other abundant architectures include two different gene arrangements leading either to a polyUb precursor protein as the result of a spacer-less tandem fusion of several Ub monomers (42% of all architectures) or to a single Ub protein fused to a distinct protein (Figure 2). Regarding polyUb tandem genes, the most frequent architecture corresponds to that of a dimer (12% of all architectures), followed by a tetramer (10%) (Figure 2). In the particular case of *S. cerevisiae*, but also in mammals, Ub is encoded by four different genes (Appendix A) [31]. The *UBI4* gene encodes a polyUb precursor protein consisting of five tandem head-to-tail Ub repeats. In the case of humans, the equivalent of the *UBI4* gene corresponds to the *UBB* and *UBC* genes, which encode poly-Ub precursors consisting of three and nine tandem Ub moieties, respectively. The number of Ub coding repeats is usually below 10 in most eukaryotes; indeed, architectures containing more than nine Ub repeats are very rare (less than 2% of all architectures) (Appendix A). However, in some organisms, such as the protozoa *Trypanosoma cruzi*, a polyUb gene with more than 40 tandemly linked Ub coding sequences has been described [32]. From our analysis, we found that the longest polyUb fusion (46 Ub domains in tandem) in PFAM appears in the cichlid *Astatotilapia calliptera* (Appendix A). In addition to tandem repeats, there are examples of exceptional hybrid architectures where the tandem Ub moieties are also combined with other domains (Appendix A).

(iii) Ub can be fused to other proteins. Strikingly, Ub is almost exclusively fused to the eS31 and eL40 ribosomal proteins (Figure 2), although fusions with other proteins can be identified in different eukaryotes (Appendix A). Normally, a single Ub moiety, which is positioned as the N-terminal part of the precursor protein, is fused to a single eS31 or eL40 ribosomal protein; however, Ub-eS31 or Ub-eL40 fusions combined in other different architectures can also be found (Appendix A). Curiously, the combinatorial capability of the Ub-eS31 fusion with other domains is much more restricted than that of the Ub-eL40 fusion (for examples, see Appendix A). In the particular case of *S. cerevisiae* (see Appendix A), the paralogous *UBI1* and *UBI2* genes (human *UBA52* gene) encode a single Ub moiety fused to the 60S ribosomal proteins eL40A and eL40B, respectively. The *UBI3* gene (human *RPS27A/UBA80* gene) encodes a single copy of Ub fused to the 40S ribosomal protein eS31 [31,33]. Although exceptions have been identified, it can be generally said that, in most eukaryotes, the amino acid sequence of the Ub proteins originating from the different genes are identical, suggesting that the different loci have undergone concerted evolution, which homogenizes their sequences by gene conversion [34]. Our analysis confirms that the Ub-eL40 and Ub-eS31 fusions are universally distributed amongst model eukaryotes (Figure 2 and Appendix A). Only rarely, Ub is fused to other proteins in different architectures (Appendix A), although these arrangements are widely represented in the eukaryotic groups of animals, fungi, and plants (Figure 3). In this sense, in a small group of mixotrophic algae, it has been reported that one or several N-terminal Ub moieties are fused in-frame inter alia to ribosomal protein P1, actin, a zinc-finger protein, a nickel superoxide dismutase, or a protein with similarity to bacterial integral membrane proteins [35,36]. A more careful study indicates that the Ub fusion to the latter three proteins is found in other eukaryotes, such as different types of algae and diverse fungi [36]. Indeed, our analysis shows that the zf-AN1 (PF01428) and zf-RRN7 (PF11781) domains, which correspond to zinc-finger domains, are the most abundant ones in these unusual architectures (ca. 1.2% abundance). We have also identified a few cases of Ub fusions with other ribosomal proteins different from eS31 or eL40, such as eS8, eS19, uL1, or eL41 (Appendix A). Interestingly, in some cases, the domain arrangement in the fusion proteins is different to the “classical” N-terminal position of Ub, as the Ub moiety can also occupy a C-terminal or even an internal position relative to its fusion partner (e.g., [36], see also Appendix A). 

Free monoUb is generated by proteolytic cleavage of the last C-terminal residues (single Ub gene) or at the Ub-protein junction (Ub fused genes) by specific proteases or deubiquitinases (e.g., Appendix A) [37], an aspect that has, however, only been experimentally validated in few organisms. In addition, given the large redundancy found for these enzymes in model eukaryotes, the responsible deubiquitinase(s) could so far not be unambiguously revealed. In yeast, human, and other model eukaryotes, no intact Ub precursors can be detected unless cleavage-retarding or -inhibitory mutations were introduced at the C-terminus of the Ub moiety in the precursors (e.g., [38,39]); this indicates that processing is a fast, likely co-translational, event. An interesting in vitro rabbit reticulocyte lysate-based translation system has been developed to tackle Ub maturation by deubiquitinases [40]. The results obtained by this system have allowed suggesting that processing of human UBA52 and UBA80 precursors occurs mostly post-translationally while that of UBB or UBC precursors probably occurs through a combination of co- and post-translational mechanisms [40]; however, we believe that these results are far from the real physiological situation and therefore must be validated by in vivo studies. Further biochemical analyses upon fractionation of a mouse liver cytosolic extract have allowed the identification of distinct deubiquitinases as the enzymes responsible for the processing of either human Ub-eL40, Ub-eS31, or polyUb precursors [40], but again, these experiments do not exactly reflect the in vivo situation. One notable exception to proteolytic maturation of a Ub precursor has been found in *G. lamblia*; in this organism, the Ub-fused to eS31 is not cleaved, presumably because an additional alanine is present between the two glycine residues at the junction between the Ub-like domain and the eS31 protein [41].

## 3. The Significance of Ubiquitin Fusion Genes

Why evolution has selected unusual gene fusions to produce de novo Ub by proteolytic cleavage of precursor proteins in practically all eukaryotes is a mystery. It has been speculated that a polyUb precursor consisting of tandem head-to-tail Ub repeats could allow the prompt synthesis of large amounts of Ub as a cellular response to special environmental conditions, such as a variety of sudden insults [42]. In this sense, the yeast Ubi4 precursor is the main source of Ub when cells enter the stationary phase or when they are subjected to different stress conditions, including high temperature, oxidative stress, or starvation [42,43]. In agreement with this, a *ubi4∆* yeast strain is not only hypersensitive to different stresses, but also prematurely induces apoptosis and shows decreased replicative lifespan [44]. Interestingly, a recent study shows that, in laboratory and industrial yeasts, the variation in the number of Ub moieties derived from a polyUb gene modulates Ub-dependent proteasome activity following a heat shock and suggests a positive correlation between the number of Ub moieties and cell survival, with different repeat numbers being optimal for coping with different stress conditions [45]. Similar conclusions can be obtained from the interpretation of the transcriptional regulation of polyUb genes in different animals and plants (e.g., [46,47,48]) or from the consequences of the deletion of the mouse *UBC* gene for proliferation and stress tolerance of embryogenic fibroblasts during fetal development [49].

As mentioned above, Ub is also expressed from fusion genes containing a single Ub coding unit combined in frame with the coding region of another protein that is unrelated to Ub. In a vast majority of eukaryotes, these proteins are the ribosomal proteins eS31 and eL40. In microorganisms, such as yeast, most cellular Ub originates from the *UBI1*, *UBI2*, and *UBI3* genes under physiological growth conditions [31,43]. This fact invites us to consider the attractive idea that fusing the Ub moiety to a ribosomal protein could have evolved to couple the synthesis and degradation of proteins to maintain proteostasis in eukaryotes, this way explaining the selective advantage of having fused Ub to either eL40 and eS31 in most eukaryotes and even to eS30, eS19, eL41, or P1 in those organisms where these fusions have arisen. However, this reasoning is unable to sufficiently explain why eL40 and eS31, apart from the latter anecdotic fusions to other ribosomal proteins, have been specifically selected in eukaryotes and suggests a precise and important role for Ub in eS31 and eL40 expression, ribosome assembly, or ribosome function, or, vice versa, a specific relevance of these ribosomal proteins for the fused Ub. 

(i) However, the Ub moiety of Ub-fused eL40 and eS31 is not strictly required for the function of these two ribosomal proteins. Examples of stand-alone genes encoding either eS31 or eL40 without N-terminal Ub fusions can naturally be found in a large variety of organisms, mainly archaea but also algae, plants, fungi, or animals, when searching the eS31 and eL40 domains (PF01599 and PF01020, respectively) in the protein family database PFAM [27] (Appendix A). Moreover, as demonstrated in yeast, constructs expressing eL40 and eS31 proteins lacking their N-terminal Ub moieties can fully complement *ubi1*, *ubi2*, and *ubi3* null mutants, respectively; nevertheless, this is only possible as long as these unfused constructs are overexpressed from high-copy-number plasmids, suggesting that indeed the N-terminal ubiquitin moiety of the Ubi1 and Ubi3 precursors contributes to the efficient expression (synthesis and/or folding) of eL40 and eS31, respectively, or to their assembly into ribosomes. However, when the sole cellular source of eL40 or eS31 originates from a single-copy allele of *UBI1* or *UBI3* lacking the Ub-coding sequence (*ubi1∆ub* and *ubi3∆ub* alleles, respectively), integrated at the native loci, cells showed a pronounced slow-growth phenotype due to the shortage of the corresponding ribosomal proteins and respective ribosomal subunits [33,38,50]. Due to these observations, it has been suggested that the N-terminal Ub moiety fused to eL40 and eS31 could act as a cis-acting chaperone to facilitate the correct folding and hence the efficient production and accumulation of these ribosomal proteins [33,50], thus performing what Varshavsky and co-workers had proposed to be the primordial Ub function [51]. In agreement with this hypothesis, a minor but significant aggregation of a C-terminally HA-tagged eS31 protein could be observed when it was produced from a Ub-free Ubi3∆ub-HA precursor, a tendency that was, however, not observed when eL40A-HA was generated from the *ubi1∆ub*-HA allele [50]. Moreover, the replacement of the Ub moiety of Ubi1 by the yeast Ub-like SUMO protein (yeast Smt3, see below), which has been proven as an N-terminal fusion partner to augment the production of recombinant heterologous proteins through significant improvement of protein stability and solubility (e.g., [52,53]), was able to modestly increase the steady-state levels of the mature eL40A-HA protein and, therefore, to slightly but significantly suppress the growth defect of a *ubi1∆ub ubi2∆* mutant strain [50]. Thus, altogether, these observations provide adequate experimental evidence to suggest that Ub (or Smt3) has indeed a minor but positive role as a cis-acting chaperone for the correct folding of eL40 and eS31 [50]. Accordingly, it has been directly demonstrated that heterologous gene expression in yeast can be considerably increased by expressing particular proteins as Ub fusions [24]. This function is unlikely to occur in trans, as the expression of free Ub molecules in *ubi1∆ub ubi2∆* or *ubi3∆ub* cells did not result in the suppression of the growth defects of these mutants ([50] and S. M.-V., unpublished results). 

(ii) In addition to the specific function for Ub in eL40 and eS31 expression, we have also addressed whether the Ub moiety of the yeast Ubi1 and Ubi3 precursors has a role in the assembly and function of the respective eL40A and eS31 ribosomal proteins. We considered this question to be very pertinent, especially given the strategical location of eL40 and eS31 on both sides of the binding site of translational GTPases (Figure 4). First, different experimental evidence suggests that the Ub moiety of Ubi1 and Ubi3 seems not to directly contribute to the assembly of the respective eL40A and eS31 proteins. Thus, under wild-type conditions, the Ubi1/2 and Ubi3 precursor proteins have so far never been detected, indicating a very rapid proteolytic maturation. Therefore, it is very improbable that the Ub molecule fused to eS31 may directly participate in the assembly of this ribosomal protein into pre-40S ribosomal particles as this occurs in the nucleus [54,55]. The ribosomal protein eL40 assembles in the cytoplasm [56]; thus, a direct role of its fused Ub molecule could be theoretically possible but would require that this should practically happen in an almost co-translational manner. Still, it is possible that the Ub moiety of a Ubi1/2 or Ubi3 precursor forms a non-covalently linked complex with the respective ribosomal proteins once cleaved; thus, these Ub molecules would operate as a kind of dedicated chaperone for the assembly of these ribosomal proteins. Interestingly, such a molecular complex between Ub and eL40 has been suggested to form upon cleavage of the mouse UBA52 precursor [39], but whether Ub is acting as a dedicated chaperone for eL40 assembly or eL40 is working as a carrier to bring Ub to the nascent 60S ribosomal subunits has not yet been elucidated. It is now clear that, at least in yeast, the presence of Ub obstructs the assembly of eL40 or eS31 into pre-ribosomal particles [38,57]. While cleavage-resistant Ubi1 or Ubi3 precursor variants, generated by mutating the terminal region of Ub to partially or totally impair Ub removal by specific deubiquitinases, are able to incorporate into nascent pre-ribosomal particles, this occurs much less efficiently than in the case of the processed eL40 or eS31 proteins derived from wild-type Ubi1 or Ubi3 precursors, respectively; consequently, non-cleaved Ubi1 and Ubi3 are also highly instable [38,57]. Besides being due to the generally observed rapid degradation of unassembled ribosomal proteins (see next section), degradation of these precursors could also be specifically facilitated by the fact that a non-cleavable Ub moiety, when fused to the N-terminus of another protein, can function as a degradation signal in the so-called Ub fusion degradation pathway, as demonstrated for artificially engineered, non-cleavable Ub fusion proteins [58,59]. Forcing the assembly of non-cleaved Ubi1 and Ubi3 proteins has important functional consequences. The incorporation of Ubi3 into nascent 40S ribosomal subunits does not impair their biogenesis but leads to translation initiation defects and hypersensitivity to antibiotics targeting translation [38]. Strikingly, as mentioned above, a natural non-cleaved form of Ubi3 is present in all ribosomes from *G. lamblia* without apparent functional consequences [41]. In turn, the incorporation of Ubi1 into 60S ribosomal subunits affects their biogenesis, due to the assembly factor Tif6 not being properly recycled, and impairs their function during translation elongation; most likely, these defects are the consequence of the interference of the unprocessed Ubi1 with the binding and function of the GTPases that bind to the GTPase-associated center of the ribosome, such as Efl1, involved in the recycling of Tif6, and eEF1A and eEF2, which are general translation elongation factors [57]. 

The biological significance of Ub fusions to other proteins, including actin, nickel superoxide dismutase, zinc-finger-containing proteins, or proteins with similarity to bacterial integral membrane proteins, is, if there is any, even more intriguing and currently unknown.

## 4. Other Roles of Ubiquitin during Ribosome Biogenesis and Function

In addition to the above-discussed function of Ub as a facilitator of folding, stability, and assembly of the two ribosomal proteins eL40 and eS31, many other important roles have been described for this molecule in the processes of ribosome biogenesis and translation. In these cases, ubiquitination may have antagonistic outcomes, either a destabilizing role by targeting ribosomal components to proteasomal degradation or a stabilizing regulatory role important for ribosome biogenesis or function [18]. Even if ubiquitination would work towards protein degradation, its role during ribosome biogenesis and translation appears to be critical, as suggested by the finding that inhibition of proteasome activity generates multiple defects along these two pathways [60]. Crucial aspects of Ub-mediated mechanisms during the biogenesis [61] and function [7] of ribosomes have been recently reviewed. 

(i) It has been well-known for decades that, both in yeast and mammals, ribosomal proteins are rapidly degraded when they are not assembled into functional ribosomal subunits owing to impaired pre-rRNA synthesis or processing [62,63,64], or as the result of the attempted overexpression of ribosomal proteins from high-copy-number plasmids [65,66,67]. This phenomenon suggested the existence of a quality control mechanism that ensures that excess free ribosomal proteins, which generally contain highly basic and intrinsically disordered extensions, do not accumulate, thereby preventing their aggregation or inappropriate interaction with other cellular components [68]. In mammalian cells, the dynamic behavior of nucleolar proteins has been studied by fluorescence microscopy and quantitative proteomics, showing that newly synthesized ribosomal proteins quickly accumulate in the nucleolus, but then a significant fraction of them is not assembled into ribosomal subunits and continuously degraded, most likely by proteasomes located in the nucleoplasm [69]. This observation reflects that the stoichiometric production of all the required ribosomal components (rRNA and ribosomal proteins) needed to ensure efficient ribosome assembly is reached through an expensive mechanism consisting in the excess supply of ribosomal proteins. Similarly, in yeast, unassembled ribosomal proteins, owing to their excess production, are rapidly ubiquitinated and degraded by the proteasome, again mainly in the nucleus (e.g., [70]). Further experiments have defined how this degradation process operates and how it permits to distinguish between assembled and free ribosomal proteins [71]. Thus, the yeast E3 Ub ligase Tom1 and its mammalian orthologue HUWE1 have been identified as the specific Ub-conjugating enzymes that recognize and ubiquitinate free ribosomal proteins at specific lysine residues [71]. Interestingly, these lysines are no longer accessible when ribosomal proteins are bound by their dedicated chaperones (e.g., uL4 and Acl4, [72]) or have assembled into pre-ribosomal particles; thus, providing an explanation for how free ribosomal proteins are specifically selected for degradation and, thereby, prevented from succumbing aggregation.

(ii) Certain free ribosomal proteins exert a regulatory function on cell proliferation and apoptosis, which is especially important as a response to a variety of stresses that impair ribosome synthesis. In this function, Ub is again involved. Thus, in situations where mammalian ribosome biogenesis is impaired, the so-called nucleolar stress response takes place [73]; this phenomenon consists in the translocation of unassembled ribosomal proteins from the nucleolus to the nucleoplasm where selected ones can then bind the E3 Ub ligase MDM2 (human HDM2) [74]. MDM2 is known to polyubiquitinate the tumor suppressor p53 protein, thus maintaining low p53 protein levels under normal cellular conditions [75]. Under stress conditions that elicit the nucleolar stress response, p53 is stabilized and activated to induce cell cycle arrest and apoptosis [74,76,77]. Under these conditions, several free ribosomal proteins, among them uL5, which are unable to assemble into pre-ribosomal particles, accumulate in the nucleoplasm, where they bind to MDM2, thereby blocking p53 ubiquitination and proteasome-dependent degradation [78,79]. 

MDM2 can also have a role that is opposite to the one involving promoting the degradation of p53. It has been demonstrated that MDM2 extends the half-life of distinct proteins; among them is E2F1, a transcription factor that regulates the expression of different genes required for the entry and passage through the S phase of the cell cycle [80]. Binding of MDM2 to E2F1 inhibits its ubiquitination by the E3 Ub ligase SCF^SKP2^, thereby preventing E2F1 degradation by the proteasome [81]. Upon activation of the nucleolar stress response, it has been shown that binding of free ribosomal proteins (e.g., uL5) to MDM2 does not only stabilize p53, but also causes the release of E2F1, which is subsequently degraded in the nucleoplasm; as a result, the cell cycle arrests in the G1 phase [82].

In yeast, a process that coordinates ribosome integrity and cell proliferation, notably showing some parallelism to the mammalian nucleolar stress response, has been identified [83]. Thus, as a consequence of a specific inhibition of RNA polymerases I or III, yeast cells respond by delaying the G1/S transition of the cell cycle, a process that involves the accumulation of free ribosomal proteins, among them uL5 and uL18. As yeast lacks a p53 homologue, it remains to be determined how exactly the signal stemming from impaired ribosome biogenesis is transmitted to exert cell cycle control. Moreover, whether or not ubiquitination plays a role in this surveillance mechanism must still be experimentally addressed.

(iii) While free ribosomal proteins are highly unstable and efficiently degraded, pre-ribosomal particles and, especially, mature ribosomes are relatively stable entities, unless any of their relevant components or factors involved in their biogenesis is missing, not properly associated, or exhibiting reduced activity or the functionality of mature ribosomes is perturbed (for a review, see [84]). Nevertheless, under certain conditions, degradation of correctly functioning mature 40S and 60S ribosomal subunits can also be mediated, in a Ub- and proteasome-independent manner, via a specific autophagy pathway, known as ribophagy, which culminates in the engulfment of cytoplasmic ribosomes by the vacuole (in yeasts) or the lysosome (in mammalian cells) [85]. In yeast, ribophagy has special relevance during stress conditions, such as nutrient deprivation, allowing the recycling of essential cellular building blocks for cell survival. Whether ribophagy also contributes to the degradation of mature ribosomes in normal growth conditions is still unclear. In any case, Ub has also a crucial role in selective ribophagy, as both Ubp3, a deubiquitinase, and its cofactor Bre5 are required for the degradation of 60S ribosomal subunits [86]. As Cdc48, a chaperone-like AAA-type ATPase, and Ufd3, a co-factor of Cdc48 with Ub-binding activity, are also involved in this pathway [87], it has been assumed that ubiquitinated targets in the 60S ribosomal subunits are recognized first by the Cdc48-Ufd3 complex and then deubiquitinated by the Ubp3-Bre5 heterodimer before vacuolar targeting and degradation [87]. Further experiments have identified the mono-ubiquitinated K74 residue of ribosomal protein uL23 as one, but not the only, substrate of Ubp3 in this process [88]. Mono-ubiquitination of assembled uL23 at its K74 residue involves the E3 ligase Ltn1 and seems to prevent ribophagy of 60S ribosomal subunits [88]; consistently, upon ribophagy induction by nitrogen starvation, Ltn1 steady-state levels are significantly reduced as both its de novo synthesis is decreased, owing to the inhibition of translation under starvation conditions, and pre-existing Ltn1 is auto-ubiquitinated and degraded by the proteasome [88]. It is also worth mentioning that Ubp3 appears not to be required for the degradation of mature 40S ribosomal subunits upon starvation; thus, how exactly 40S ribosomal subunits are channeled to ribophagy needs further clarification [84,86]. Whether or not Ub is required for ribophagy in mammalian cells is still unknown [89].

(iv) Ribosomal proteins are also subjected to ubiquitination upon other cellular insults than those caused by nutrient starvation. Thus, for instance, Ub seems to have a regulatory role during the induction of the unfolded protein response (UPR) [90]. Higgins and co-workers have shown that, upon the stimulation of UPR or inhibition of translation in human cells, there is a rapid cytoplasmic mono-ubiquitination of specific lysine residues of distinct ribosomal proteins belonging to the mature 40S ribosomal subunit, such as uS5, uS3, uS10, and RACK1 [90], which are all located on the solvent-exposed surface of the subunit [91]. The phenomenon is well-conserved between different eukaryotes, from yeast to human cells. Importantly, these Ub modifications must be functionally relevant as a failure to ubiquitinate the specific lysine residues on the above ribosomal proteins (e.g., by mutation of these residues) enhances the sensitivity to drugs that lead to sustained UPR activation and cell death [90]. However, the exact significance of this UPR-induced ubiquitination of assembled 40S subunit ribosomal proteins remains unclear; in other words, in which sense the function of ubiquitinated ribosomes differs from the one of non-modified ones during the execution of the UPR program is still a mystery. 

(v) Functionally defective cytoplasmic ribosomes harboring point mutations in relevant rRNA sites (e.g., residues in the 18S rRNA at the decoding center and residues in the 25S rRNA at the peptidyl transferase center) do not engage in translation; instead, they are actively eliminated by a mechanism named non-functional rRNA decay (NRD), which was revealed in yeast [92]. As it is unlikely that both ribosomal subunits are simultaneously defective in nature, evolution has selected distinct mechanisms to specifically eliminate either faulty large or small subunits. In the case of defective 40S ribosomal subunits, this process involves the sequential mono- and poly-ubiquitination of ribosomal protein uS3 at specific lysine residues by E3 Ub ligases: first by Mag2 (mono-ubiquitination) and then by Hel2 and Rsp5 (poly-ubiquitination). Upon dissociation of the 80S ribosome, the defective 40S ribosomal subunit is degraded by a process that requires the exonuclease Xrn1 but, apparently, does not involve the Ub-dependent proteasome [93,94]. Recognition of defective 40S subunits seems to occur via RACK1, which interacts with the C-terminal region of uS3 [93]. In the case of the 60S ribosomal subunit, the NRD process involves ubiquitination of one or several so far unidentified ribosomal protein(s) of the large ribosomal subunit, mediated by the Rtt101-Mms1 E3 Ub-ligase complex, as well as dissociation of the defective 60S subunit from the 80S ribosome, which is dependent on the Cdc48-Npl4-Ufd1 complex [95,96]. Ub-modified 60S ribosomal subunits seem to then be transferred to the proteasome for protein degradation; partially dismantled 60S subunits are then accessible to RNases, which degrade the mutated 25S rRNA [96]. However, how defective 60S ribosomal subunits are actually efficiently recognized, Ub-targeted, and degraded is still unknown [7]. Moreover, it has been proposed that this process is not only involved in the clearance of non-functional 60S ribosomal subunits but could also continuously operates to adjust the steady-state levels of 60S ribosomal subunits in normally growing cells [96]. 

(vi) Ubiquitination is also implicated in additional situations, likely again as an adaptive response to deal with intracellular or environmental insults, as, for example, those leading to ribosome stalling on mRNAs. In these circumstances, ubiquitination targets ribosomal proteins and involves factors that are shared with those previously described to participate in the NRD and UPR processes, such as RACK1, uS10, Hel2 (ZNF598), or Ltn1 [97]. Another example of a process where ubiquitination plays a role is the response to oxidative stress, which involves K63-linked poly-Ub chain formation on numerous ribosomal proteins of both ribosomal subunits. In a pioneer work, Spence et al. demonstrated that the ribosomal protein uL15 is K63 poly-ubiquitinated in yeast and mammals during the S phase of the cell cycle or upon impairing translation by the use of translation inhibitors [98]. However, uL15 is not the sole K63 Ub-modified ribosomal protein in these conditions. Instead, many other ribosomal proteins from both subunits also experience K63 poly-ubiquitination, as observed in response to oxidative stress in yeast; interestingly, several modified ribosomal proteins of the 40S subunit cluster in or around the head domain (uS5, uS3, eS12, uS10, eS21, and RACK1) [99,100]. Comparison of the cryo-EM structures of ribosomes harboring or lacking K63 Ub has suggested that this modification regulates translation at the level of the elongation phase [101]. Thus, K63 poly-ubiquitination of specific ribosomal proteins, including the P-stalk protein uL10, has been suggested to prevent the binding of the elongation factor eEF2 and impair translation, providing a remarkable structural basis for the function of this kind of Ub modification [101].

## 5. Ubiquitin-Like Modifiers Related to Ribosome Biogenesis

In this section, we provide an overview of the Ub-like proteins that have been connected to the biogenesis and function of ribosomes. We will focus on those identified in *S. cerevisiae* and mention only briefly those that are not present in this model microorganism. As Ub, all these Ub-like proteins are used to covalently modify other molecules, mainly proteins. 

(i) SUMO, which in yeast is known as Smt3, is encoded by a single essential gene [102,103]. Smt3, as its orthologues in higher eukaryotes, is a protein of about 100 amino acids that shares rather discrete sequence (about 20%) but high structural identity with Ub (Figure 1). However, all SUMO proteins, including Smt3, contain a flexible N-terminal extension of around 10 to 25 amino acids that is absent in Ub and participates in the formation of SUMO chains as it comprises some conserved lysines (K11, K15, and K19) for conjugation [104,105].

Smt3 is synthesized as a precursor protein that is extended by three additional amino acids at its C-terminal end; thus, these residues, which are situated after the invariant di-glycine motif defining the C-terminal end of the mature protein, must be removed by specific proteases to activate Smt3 for conjugation to its substrates [105,106]. Then, the activated Smt3 is covalently attached by the formation of an isopeptide bond between the carboxyl group of its C-terminal glycine and the epsilon-amino group of a lysine residue in the acceptor protein. The sumoylation process, which is reversible, requires a cascade of steps that involves enzymes (E1 to E3 enzymes) functionally related to those used for ubiquitination [105]. As Ub, Smt3 has hundreds of protein substrates and sumoylation leads to changes in the expression, solubility, stability, and protein interaction properties of these substrate proteins, which participate in multiple cellular functions, such as DNA and RNA metabolism, cell-cycle progression, and membrane or organelle dynamics [105]. Specifically, more than 200 different substrates have been identified in yeast in normal growth conditions (e.g., [107,108]). It has been shown that both the number and pattern of sumoylated substrates significantly increase when cells are subjected to changes in growth media and conditions or following various cellular stresses [109,110].

Smt3 has an important role in yeast ribosome biogenesis too; thus, impairing the sumoylation pathway leads to defects in pre-rRNA processing and 60S ribosomal subunit maturation and export. Moreover, Smt3 reversibly modulates the function and localization of several factors involved in the biogenesis of both ribosomal subunits [111], a role that seems to be conserved in mammals (e.g., [112]). The relevance of the Smt3 modifications in the different substrates is unclear as mutations of Smt3 acceptor sites within individual ribosome biogenesis factors or ribosomal proteins do not lead to any observable phenotype (discussed in [111]).

(ii) Rub1, the orthologue of mammalian NEDD8, is a yeast protein of 77 amino acids with sequence and structural homology to Ub [113]. As other Ub-like proteins, Rub1 is also synthesized as a precursor protein, which includes an additional C-terminal asparagine residue following the characteristic di-glycine motif. The activation of the GG dipeptide requires the proteolytic removal of this asparagine to allow productive conjugation of Rub1 onto a well-conserved lysine residue within any of its substrates; this reaction, known as neddylation, occurs by an analogous pathway to that described for Ub conjugation or ubiquitination [113]. In contrast to Ub and Smt3, Rub1 is not essential for yeast growth, although *RUB1* deletion confers hypersensitivity to drugs interfering with DNA replication and repair [114]. The most well-characterized substrates of Rub1/NEDD8 are cullins, which are structural components of specific E3 Ub ligase complexes and, in yeast, encompass only the Cdc53, Cul3, and Rtt101/Cul8 protein [114]. The neddylation of cullins stimulates the activity of these ligases, thus increasing ubiquitination efficiency and, as a consequence, the Ub-dependent proteolysis of their substrate proteins [21,115]. Proteins unrelated to cullins have also been identified as targets of Rub1/NEDD8. Of special interest for this review is the fact that almost half of all ribosomal proteins are neddylation substrates in mammalian cells [116]. Notably, neddylation regulates the stability of ribosomal proteins in the nucle(ol)us. Thus, among other activities, neddylation appears to protect ribosomal proteins from nucleoplasmic Ub-dependent proteasomal degradation, likely ensuring their nucleolar localization [116]. In mammals, neddylation of ribosomal proteins seems to be carried out by the E3 Ub ligase MDM2 [78]. As described above, MDM2 polyubiquitinates p53 to maintain it at low levels in normal cellular conditions [75]; however, when the nucleolar stress response is induced, p53 is stabilized and activated to trigger a cell cycle arrest [74,76,77]. As also already mentioned above, several free ribosomal proteins, among them uL5, translocate, when not efficiently assembled into pre-ribosomal particles, from the nucleolus to the nucleoplasm, where they bind to MDM2 and block ubiquitination and the proteasome-dependent degradation of p53. Accumulation of uL5 in the nucleoplasm seems to be enabled by its rapid de-neddylation [78,79]. Simultaneously, the nucleoplasmic redistribution of non-neddylated uL5 promotes its binding to c-Myc, thus lowering the activity of this central proliferation-stimulating transcriptional factor [117], as well as reducing its steady-state levels. Another ribosomal protein that is neddylated and also enables, by binding to MDM2 and inhibiting MDM2-mediated ubiquitination, stabilization and activation of p53 is uS11 [118]. Neddylation of free uS11 has been described to increase the association of uS11 with its binding partner hCINAP (Fap7 in yeast) [119]. Then, in a regulatory feedback loop, uS11-hCINAP recruits a specific NEDD8 protease, which reduces uS11 neddylation and, as a consequence, decreases uS11 stability and changes its localization pattern [119]. 

Whether or not an analogous Rub1-dependent neddylation of yeast ribosomal proteins occurs, and if it would have any impact on cell proliferation and cell cycle arrest, is still unknown.

(iii) Additional yeast Ub-like modifiers comprise the autophagy related Atg8 (LC3 in mammals) and Atg12 (ATG12 in mammals) proteins, which both work in the same step during autophagy, consisting of the conjugation of Atg8 to the membrane lipid phosphoethanolamine [21,120,121]. Neither of these two Ub-like proteins has been described to participate in ribosome biogenesis and function so far, although it has recently been shown that ATG12 supports the translation of a subset of mRNAs involved in cell cycle control and DNA repair in mammalian cells [121]. Moreover, yeast ribophagy (see above) seems to be dependent on the Atg8 conjugation system [86]. 

The non-essential Ub-like modification Urm1 has been described to act as a sulphur carrier that contributes to the fine-tuning of translation by enabling the chemical thiolation of tRNAs at their wobble uridines in the cytoplasm [122].

Finally, a non-conventional Ub-like protein has been identified and called homologous to UBiquitin 1 (Hub1 in yeast, UBL5 in mammals). Unlike Ub and other Ub-like proteins described herein, Hub1/UBL5 does not possess a characteristic di-glycine motif at its C-terminal end. Instead, there is a conserved di-tyrosine followed by a single amino acid that in yeast is leucine [123]. Hub1 has been shown to modify some proteins related to cell polarity and to have an important function during splicing [124]. Whether Hub1 is conjugated to its targets or is non-covalently associated with these is still controversial (discussed in [125]).

(iv) Moreover, three additional Ub-like modifiers (ISG15, FAT10, and UFM1) have so far been identified and studied; however, these have no yeast counterparts. Notably, both ISG15 and FAT10 are made up of two Ub-like domains that are connected by a short flexible linker [21]. Although not related to ribosome biogenesis, ISG15 has been shown to be co-translationally conjugated to a set of newly synthesized proteins via its predominant, polysome-associated E3 ligase HERC5; in interferon-stimulated cells, these proteins are primarily viral proteins, although not only proteins related to antiviral immunity have been identified among ISG15 substrates [126]. 

FAT10, which is only found in mammals, also appears to be conjugated, although not exclusively, to nascent proteins [127,128]. FAT10 has been reported to perform diverse roles within the immune system and/or inflammation processes in other tissues; it presents a discrete set of targets for its conjugation, among them misfolded proteins, monomer proteins unable to assemble into their macromolecular complexes, and proteins with a very short half-life due to their narrow temporal functional requirements [128]. Interestingly, FAT10 is the only Ub-like modifier that, akin to Ub, is able to target proteins for proteasomal degradation. However, FAT10 seems to be degraded along with its substrates, therefore making the existence of FAT10-specific deconjugating enzymes unnecessary, unless this modification also harbors a not yet discovered regulatory role [128,129].

The metazoan-specific UFM1 protein is a Ub-like modifier whose function has been recently linked to the ribosome-assembled ribosomal protein uL24 [130]. This ribosomal protein is located adjacent to the ribosomal exit tunnel close to the docking site of the signal recognition particle and the translocon [91]. UFMylation of uL24 seems to have a role in the co-translational translocation of proteins into the endoplasmic reticulum [130].

(v) Finally, another Ub-like protein, FUBI, has been identified as an N-terminal fusion to ribosomal protein eS30 in nematodes and mammals [131,132]. This fusion resembles that of Ub to eL40 and eS31, and, similarly, the FUBI moiety is removed from the fusion precursor before the assembly of eS30 into pre-40S ribosomal particles; although direct evidence is still lacking, the FUBI protein is most likely separated from eS30 in the cytosol by post-translational cleavage of the precursor protein [131,132]. While the FUBI protein has only limited identity to Ub, it retains a C-terminal di-glycine signature after its proteolytic separation from eS30 and could theoretically be conjugated onto target proteins; however, FUBI modification of substrate proteins has so far not been reported. Moreover, FUBI lacks the typical lysine residues that participate in Ub’s polyubiquitination; thus, the function of FUBI and Ub must be clearly different. Work carried out with mammalian cells has indicated a pro-apoptotic activity of FUBI; however, the exact function of FUBI and the relevance of its expression as a fusion protein needs to be further investigated [133].

## 6. Ribosome Assembly Factors Containing Ub-Like Domains

As mentioned in the Introduction, there are proteins that harbor Ub-like domains in their primary sequence. These domains are not proteolytically processed to release the Ub-like moiety and are therefore not conjugated onto other proteins. As we discuss later, these domains function as specific protein–protein interaction modules, but whether or not they could have further functions remains to be explored.

We have analyzed the *S. cerevisiae* genome for proteins containing Ub-like domains (clan Ubiquitin, CL0072) using PFAM [27]. This analysis resulted in 38 different candidate proteins (Appendix A), but only for two of these, Rsa4 and Ytm1, is an involvement in ribosome biogenesis known [134,135]. An additional two Ub-like domain containing proteins, the GTPases Rbg1 and Rbg2, seem to have a translation-related function [136].

Rsa4 and Ytm1 are essential proteins in yeast and have been shown to be structurally and functionally conserved in higher eukaryotes (NLE1 and WDR12, respectively) [137,138]. The Ub-like domains of Rsa4/NLE1 and Ytm1/WDR12 are located, as is the case of most Ub-containing precursors, in the N-terminal part of these proteins. The molecular structures of both factors have been solved by X-ray crystallography (see Figure 1), revealing that the two Ub-like domains superimpose well on each other and on Ub [138,139]. Whether or not the Ub-like domain contributes to the folding and solubility of the complete Rsa4 and Ytm1 proteins has not yet been explored.

Ytm1 and Rsa4 are associated with early and late pre-60S ribosomal particles, respectively; thus, their essential roles during the maturation of large ribosomal subunits are independent of each other [140,141]. In the case of Ytm1, expression of a truncated variant lacking the Ub-like domain results in a dominant negative growth phenotype [142]. In further agreement with an important function, the two Ub-like domains contain many well-conserved, essential residues [138,143], with a conserved glutamic acid (E80 in Ytm1 and E114 in Rsa4) being in each case of special relevance as its mutation also impairs growth and 60S ribosomal subunit maturation in a dominant-negative manner [141,143]. Interestingly, both Ytm1 and Rsa4 are functionally linked to the conserved AAA+-type ATPase Rea1 (also referred to as Midasin) [141,143]. Rea1 consists of a hexameric ATPase ring that is followed by a long tail, comprising an alpha-helical linker region, a flexible aspartate/glutamate-rich domain, and a C-terminal metal-ion-dependent adhesion site (MIDAS), which is homologous to the I-domain of integrins [141,144,145]. It has been unquestionably shown that binding of the MIDAS domain to Ytm1 and Rsa4 critically depends on the above-mentioned acidic residue in their Ub-like domain [141,143]. Coupling this physical connection with ATP hydrolysis drives the release of Ytm1, together with that of Erb1 and Nop7, from medium pre-60S ribosomal particles and, similarly, of Rsa4, together with that of the Rix1-Ipi1-Ipi3 complex, from later pre-60S ribosomal particles [139,143]. In both cases, the release of these factors allows pre-60S ribosomal particles to undergo specific structural rearrangements required for their proper maturation, such as ITS2-remodelling and rotation of the 5S RNP [146,147]. Hurt and co-workers have recently demonstrated how the MIDAS domain, due to the flexible features of Rea1’s tail, binds first to the hexameric ring and then engages in a stable interaction with the Ub-like domain of Rsa4 and Ytm1 [148]. ATP hydrolysis in the ATPase ring then exerts a pulling force that enables the release of Rsa4 and Ytm1 from pre-60S particles [145,148]. While the pre-60S intermediate containing Rea1 bound to Rsa4 has been determined by cryo-EM, the one displaying the Rea1-Ytm1 interaction still awaits visualization [143,148,149,150].

## 7. Concluding Remarks and Future Perspectives

It is clear that Ub and Ub-like proteins control a broad range of cellular aspects; therefore, it is not surprising that numerous regulatory connections of these protein modifiers with the strategical pathways of ribosome biogenesis, translation, and ribosome degradation have been acquired during evolution to maintain cellular homeostasis. Throughout this review, we hope to have appropriately enumerated and described the processes where these connections occur and highlighted the players involved therein.

Despite the exhaustive characterization of the structure of Ub and Ub-like proteins, the mechanisms of their conjugation reactions, and their specific partners, many details about the precise functions of these modifiers are still unclear. A list of unsolved aspects of the biology of Ub and Ub-like proteins includes:

(i) It remains enigmatic why specifically Ub, but not other Ub-like proteins, is encoded and expressed as head-to-tail repeats or fused to other proteins, which are practically invariably eS31 and eL40 in most eukaryotes. As discussed above, the likely biological logic of poly-Ub genes consists in the production of large amounts of Ub to properly modulate the clearance of (undesired) proteins by the Ub-dependent proteasome as a response to an unexpected circumstance (i.e., stress). In this sense, the existence of a single gene encoding a poly-Ub precursor has been interpreted as an evolutionary alternative for gene amplification by differentially increasing the number of copies of a particular gene [45]. More intriguing is a possible explanation concerning the evolutionary advantage of the Ub fusion within the Ub-eS31 and Ub-eL40 precursor proteins. It is reasonable to assume that these gene fusions arose from the in-frame insertion of a single Ub coding unit within the sequence encoding eS31 and eL40 of a primitive eukaryote. In consonance with this hypothesis, some archaea encode for unfused homologues of the ribosomal proteins eS31 and eL40. Archaea do not code for (canonical) Ub but nevertheless contain both a functional proteasome and other Ub-like protein modifiers [151]. However, in no case are eS31 and eL40 apparently synthesized as fusion proteins harboring an N-terminal Ub-like domain in archaea. 

However, what could be the biological reason for these gene fusions? It can be speculated that the production of precursor proteins formed by Ub and ribosomal proteins eS31 and eL40 ensures the co-regulation of two related cellular functions, protein synthesis and protein degradation; however, the same functional connection could have been guaranteed by linking Ub to different ribosomal proteins or even to ribosome biogenesis or translation factors in different eukaryotes. Thus, a selective pressure for maintaining these particular, but no other, fusions must exist in nature. It has been suggested that Ub acts as a cis-acting chaperone for the production of eS31 and eL40 [33,51], a hypothesis we have experimentally addressed and validated [38,50]. This function, however, likely precedes the assembly of these ribosomal proteins into pre-ribosomal particles, as indicated by the findings that Ub-fused precursors are rapidly, presumably co-translationally, processed and that the proteolytic cleavage of both the Ub-eS31 and Ub-eL40 precursor protein into the individual components is essential for the formation and functionality of the respective ribosomal subunits [38,57]. In agreement, the Ub moiety of the Ub-eL40 (Ubi1/2) and the Ub-eS31 (Ubi3) precursors can be deleted without interfering with ribosome biogenesis and function as long as the unfused constructs are expressed at a high dosage [33,38,50,152].

(ii) While both the components and the underlying mechanism of ubiquitination and subsequent proteasome-dependent degradation of excess ribosomal proteins are well known, at least in the yeast *S. cerevisiae*, it remains to be elucidated how Ub signaling regulates ribosome biogenesis. It is not clear how, when, and why many ribosome biogenesis factors are ubiquitinated [10,61,153]. Moreover, whether their ubiquitination is exerted to eliminate defective pre-ribosomal particles or has a regulatory role by altering the function or activity of selected factors to allow the optimal maturation of pre-ribosomal particles is still unknown.

(iii) Many ribosomal proteins are ubiquitinated to mediate the turnover of mature ribosomes (e.g., NRD and recycling of stalled ribosomes), while others are ubiquitinated as a cellular response to stress. How these latter modifications change the ribosome’s structure, have an impact on the interaction of the ribosome with other factors, or specifically alter the function of translation factors, thereby possibly allowing or impeding the translation of subsets of specific mRNAs, requires more research.

(iv) Similarly, an understanding of how SUMO/Smt3 influences ribosome assembly is still lacking. Proteome-wide approaches have identified several subunits of RNA polymerases I and III as well as many ribosome biogenesis factors involved in the assembly of both ribosomal subunits as SUMO substrates. Moreover, impairment of the SUMO conjugation or deconjugation processes negatively affects the assembly and export of pre-ribosomal particles. Again, the basic biochemical consequences of sumoylation on the interactions or activity of the modified biogenesis factor are completely elusive. Moreover, how neddylation regulates ribosomal protein stability is not understood either. 

(v) Last but not least, our understanding of the precise contribution of the N-terminal Ub-like domain of Rsa4 and Ytm1 to the function of the respective biogenesis factor is still incomplete. While it has been proven that the Ub-like domain of both factors is essential for their interaction with Rea1′s MIDAS domain, thereby promoting their release from pre-60S ribosomal particles and the progression of 60S ribosomal subunit maturation, it has not been addressed whether the Ub-like domain could also play a role for the optimal expression and stability of Rsa4 and Ytm1. It has been described that Ub-like domains not only have the capacity to confer proteasomal targeting but may also enable subsequent proteasomal degradation, especially when the protein contains exposed disordered regions [154]. Considering, however, that the Ub-like domains of Rsa4 and Ytm1 are followed by stable WD40 domains, this scenario is unlikely in the case of these two biogenesis factors.

In summary, the examples described in this review article have illustrated that Ub, Ub-like proteins, and Ub-like domains, besides carrying out many important cellular functions, specifically influence ribosome biogenesis, ribosome turnover, and translation. Future studies will undoubtedly provide valuable insights into the still unknown aspects concerning the function, mechanism, and regulation of these *ubiquitous* proteins in the diverse ribosome-related pathways in eukaryotic cells.

## Figures and Tables

**Figure 1 ijms-22-04359-f001:**
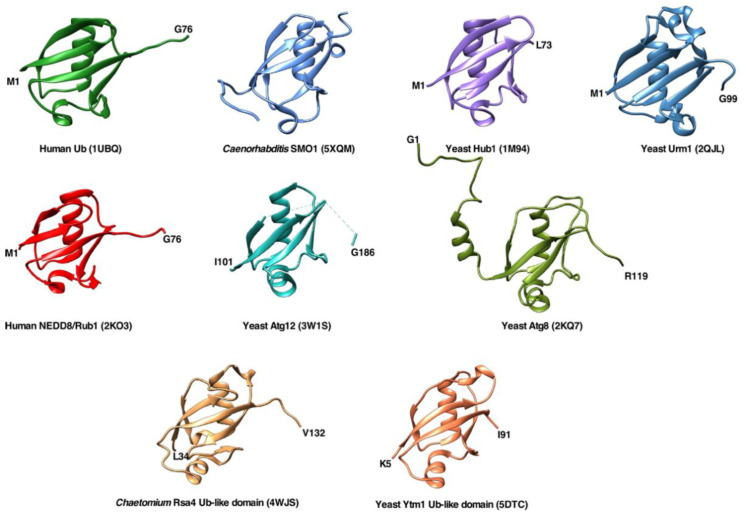
Overall structures of ubiquitin and different ubiquitin-like proteins or domains shown as ribbon diagrams. The figures were prepared using the UCSF Chimera program [4] using the structural data provided by NMR, X-ray crystallography, and cryo-EM studies deposited in the Protein Data Bank (PDB). The following structures are shown: *Homo sapiens* ubiquitin (PDB code 1UBQ), *Caenorhabditis elegans* SUMO homologue SMO-1 (PDB 5XQM), *S. cerevisiae* Hub1 (PDB 1M94), *S. cerevisiae* Urm1 (PDB 2QJL), *H. sapiens* NEDD8/Rub1 (PDB 2KO3), *S. cerevisiae* Atg12 (PDB 3W1S), *S. cerevisiae* Atg8 (PDB 2KQ7), ubiquitin-like domain of *Chaetomium thermophilum* Rsa4 (taken from PDB 4WJS), and ubiquitin-like domain of *S. cerevisiae* Ytm1 (PDB 5DTC).

**Figure 2 ijms-22-04359-f002:**
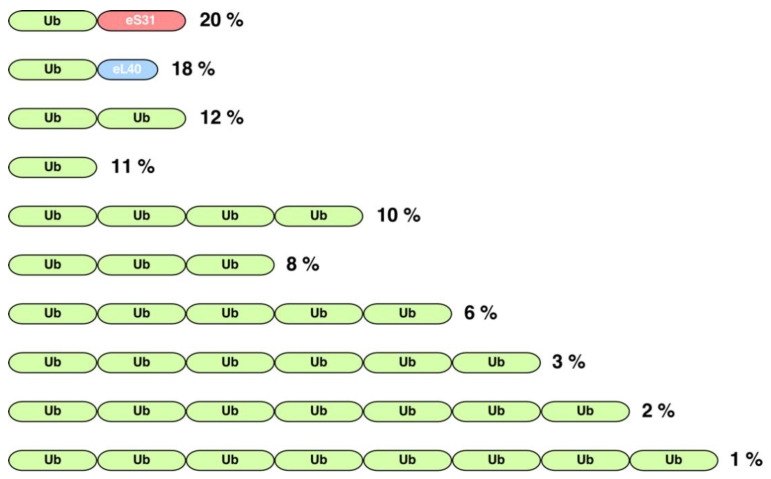
Schematic representation of the organization of the 10 most common protein architectures containing the ubiquitin domain (PFAM entry PF00240). Green blocks represent the ubiquitin sequence. Red and blue blocks represent the characteristic domains of the eS31 (PFAM PF01599) and eL40 (PFAM PF01020) family of ribosomal proteins, respectively. Note that the shown architectures comprise about 92% of all ubiquitin-containing proteins. Numbers on the right indicate the percentage of a particular architecture among all analyzed proteins. As described in the text, this analysis was carried out with all those proteins of the PF00240 entry that showed more than 80% identity over at least 80% of their length with the Ub domain of *S. cerevisiae*.

**Figure 3 ijms-22-04359-f003:**
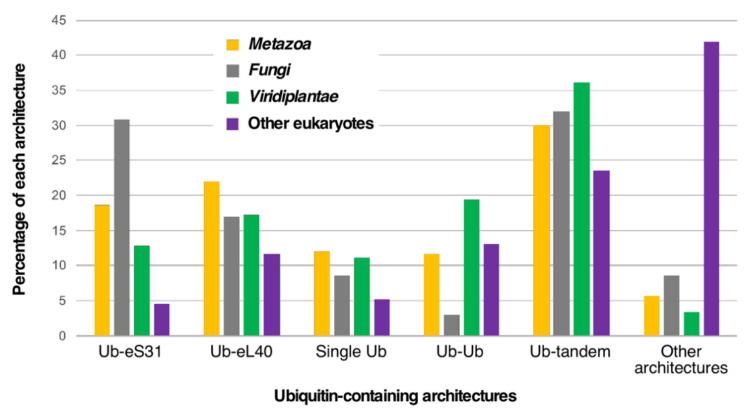
Taxonomic distribution of the most frequent ubiquitin-containing architectures among eukaryotes. Out of all possible architectures, only Ub-eS31 and Ub-eL40 fusions, monogenic Ub, dimeric Ub, and multiple Ub tandem repeats are shown. Analysis was done with different groups of eukaryotes from which we have extracted the *Metazoa* or animals, *Viridiplantae* or green algae and plants, and fungi. The percentages corresponding to each architecture category is shown for each group.

**Figure 4 ijms-22-04359-f004:**
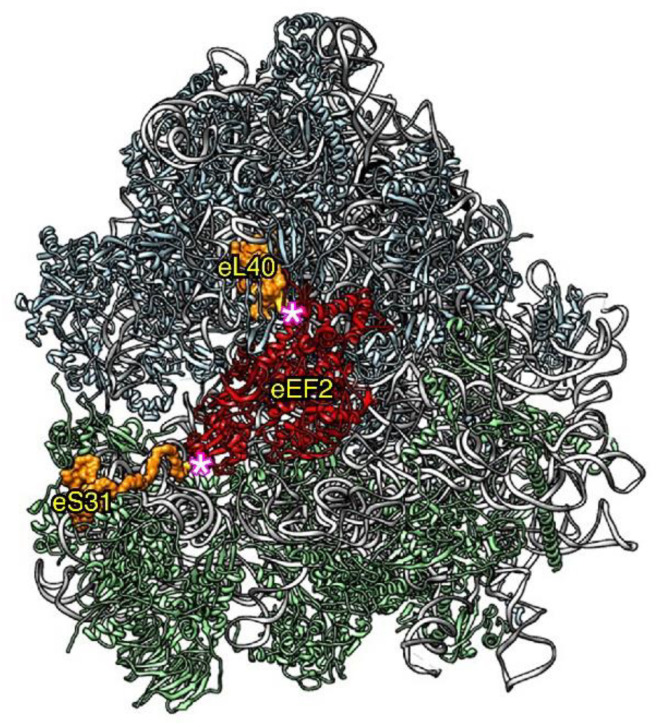
Relative position of ribosomal proteins eL40 and eS31 (orange) within the large and the small ribosomal subunit, respectively, with respect to the binding site of the GTPase eEF2 (red). The remaining ribosomal proteins are colored in blue (large subunit) or green (small subunit) and the 25S and 18S rRNA in pale gray. The 5.8S and the 5S rRNA are shown in dark gray and black, respectively. The representation was generated with the UCSF Chimera program, using the atomic model of the cryo-EM structure V of the yeast 80S ribosome bound to the Taura syndrome virus IRES (not shown) and sordarin-stalled eEF2·GDP (PDB 5JUU). Note that eL40 could be modelled from its third residue (E79) and eS31 from its sixth residue (K82); these are highlighted by a pink asterisk. Due to the positions of eL40 and eS31 within 60S and 40S ribosomal subunits, respectively, the assembly of non-cleaved Ubi1/2 or Ubi3 is expected to sterically interfere with the binding of eEF2 to the ribosomal GTPase-associated center.

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
