# Peer review of "Ubiquitin and Ubiquitin-Like Proteins and Domains in Ribosome Production and Function: Chance or Necessity?"

_ijms, 2021, doi:10.3390/ijms22094359_

Round 1

Reviewer 1 Report

This is a very thorough review of what is know about the role of ubiquitin and ubiquitin-like proteins in ribosome-production and function.  It is quite dense and not for the casual reader but for someone looking for virtually every paper published on this topic since the realization that ubiquitin can be encoded as fusion genes with ribosomal proteins and that ribosomal proteins can be post-gnomically modified by ubiquitin this manuscript will be a valuable resource.  I would liked to have come away from the review with a better mechanistic understanding of the role of ubiquitin and ubiquitin-like proteins in ribosome synthesis and function and although this represents the current state of the field, I would like to have seen the authors use their expertise to create more of a synthesis of what is known and even if speculative, provide some additional mechanistic insights on the topic.

Author Response

REQUEST: This is a very thorough review of what is know about the role of ubiquitin and ubiquitin-like proteins in ribosome-production and function. It is quite dense and not for the casual reader but for someone looking for virtually every paper published on this topic since the realization that ubiquitin can be encoded as fusion genes with ribosomal proteins and that ribosomal proteins can be post-gnomically modified by ubiquitin this manuscript will be a valuable resource. I would liked to have come away from the review with a better mechanistic understanding of the role of ubiquitin and ubiquitin-like proteins in ribosome synthesis and function and although this represents the current state of the field, I would like to have seen the authors use their expertise to create more of a synthesis of what is known and even if speculative, provide some additional mechanistic insights on the topic.

ANSWER: We appreciate that reviewer #1 considers our manuscript to be a valuable resource for people looking for publications on the topic we have reviewed. However, we also feel disappointed that she/he would have expected more from our review. As she/he indicates, we have focused on providing the current state of the knowledge about the different ubiquitin-like proteins (including ubiquitin) involved in different aspects of ribosome biogenesis and function. In addition, as initially proposed to the guest editor when we agreed to contribute to the special issue of IJMS, we have deliberately posed the question of why ubiquitin is found fused to specific ribosomal proteins (eS31 and eL40) in most eukaryotes. More importantly, we have discussed and speculated on what could be the role of ubiquitin within fusion proteins based on the available literature. As the experimental evidence is still limited and somehow controversial, we unfortunately could not provide the mechanistic insights she/he wished; moreover, we preferred not to speculate too much.

Reviewer 2 Report

Martin-Villanueva et al. present a generally well written and well-organized review of ubiquitin and ubiquitin-like protein with emphasis on their contributions to ribosome biology. The work is comprehensive and covers current knowledge well.

The following are only minor comments that reflect passages that are somewhat confusing while reading the article.

L. 49/50: ‘Accordingly, Ub is … that modifies as large number of different proteins’. As written, this sentence suggests that ubiquitin has catalytic activity and actively modifies other proteins, which is not the case. Ub is a modification. Maybe the authors can rephrase this to better make the distinction between the modifying enzymes and the modification.

L68: At least this reader would appreciate a specific example to illustrate the sentence ‘at least some types of ubiquitination still await the assignment of a specific function’. Which types?

L. 169: The description of frequencies of domain fusions is inconsistent. Here it is described as ‘it is not infrequent’. However, Table S3 is titled with ‘rare variations’. Not infrequent is not rare. Maybe it would already be sufficient to not reference S3 in the first sentence as this table is described in more detail in the next sentence. But then in L181, Table S2 is referenced as ‘only rarely’.

L. 197: Starting a new paragraph/topic with ‘in all cases’ is potentially confusing as it can connect to the previous paragraph. For example: do you mean with ‘in all cases’ that even ub in internal positions (that you just described) are cleaved by proteolytic cleavage? If this is not intended, maybe rephrase to make a clearer beginning of a new topic.

Fig. 4: The yellow asterisks are very hard to see.

L. 352: This sentence does not connect well with the preceding content.

L. 426: This is a rather long sentence that is difficult to process.

L. 549: as mutations of Smt3 acceptor sites … does not lead…

L. 600: similar to L49, is Urm1 the modifier (the enzyme) or the modification

L. 772: The title of the review poses a direct question: chance or necessity? The authors could consider providing a more direct answer to that question in the last paragraph.

Table S4: Abbreviations should be defined (in this table, ERAD appears to be the only one).

Author Response

COMMENT: Martin-Villanueva et al. present a generally well written and well-organized review of ubiquitin and ubiquitin-like protein with emphasis on their contributions to ribosome biology. The work is comprehensive and covers current knowledge well.

ANSWER: We appreciate that reviewer #2 acknowledges our manuscript as a comprehensive work covering the topic well.

The following are only minor comments that reflect passages that are somewhat confusing while reading the article.

REQUEST: L. 49/50: ‘Accordingly, Ub is … that modifies as large number of different proteins’. As written, this sentence suggests that ubiquitin has catalytic activity and actively modifies other proteins, which is not the case. Ub is a modification. Maybe the authors can rephrase this to better make the distinction between the modifying enzymes and the modification.

ANSWER:  We have rephrased the sentence to:

Accordingly, Ub is a very abundant cellular protein that is used to modify a large number of different proteins in yeast (>1000) and human (>9000) cells.

REQUEST: L68: At least this reader would appreciate a specific example to illustrate the sentence ‘at least some types of ubiquitination still await the assignment of a specific function’. Which types?

ANSWER: We have rephrased the sentence to:

However, some types of ubiquitination, especially the ones containing mixed or branched heterotypic Ubs, still await the assignment of a specific function.

REQUEST: L. 169: The description of frequencies of domain fusions is inconsistent. Here it is described as ‘it is not infrequent’. However, Table S3 is titled with ‘rare variations’. Not infrequent is not rare. Maybe it would already be sufficient to not reference S3 in the first sentence as this table is described in more detail in the next sentence. But then in L181, Table S2 is referenced as ‘only rarely’.

ANSWER: To avoid inconsistences, we have rephrased the sentence to:

however, Ub-eS31 or Ub-eL40 fusions combined in other different architectures can also be found (Tables S1-S3).

But, we have left the sentence of L181 (in our manuscript, L187-188) as is.

REQUEST: L. 197: Starting a new paragraph/topic with ‘in all cases’ is potentially confusing as it can connect to the previous paragraph. For example: do you mean with ‘in all cases’ that even ub in internal positions (that you just described) are cleaved by proteolytic cleavage? If this is not intended, maybe rephrase to make a clearer beginning of a new topic.

ANSWER: Following the advice of the reviewer, we have now omitted starting the paragraph with "In all cases":

Free monoUb is generated by proteolytic cleavage of the last C-terminal residues (single Ub gene) or at the Ub-protein junction (Ub fused genes) by specific proteases or deubiquitinases.

REQUEST: Fig. 4: The yellow asterisks are very hard to see.

ANSWER: We thank the reviewer for pointing this out. We have changed the color of the asterisks to make them clearly visible.

REQUEST: L. 352: This sentence does not connect well with the preceding content.

ANSWER: We believe that this sentence connects well the message we would like to provide. Ubiquitination has an important role in ribosome biogenesis and function, regardless of the effect ubiquitin has on its targets, be it regulation of their activity or marking them for active degradation by the proteasome.

REQUEST: L. 426: This is a rather long sentence that is difficult to process.

ANSWER: We are not sure which sentence is meant here as the line numbers of the document the reviewer is handling do not exactly correspond to the ones in the submitted document. In any case, around the lines 420-440 we did not find any long sentence that is difficult to process.

REQUEST: L. 549: as mutations of Smt3 acceptor sites … does not lead…

ANSWER: This sentence starts on line 539 in our document. We have corrected the sentence to "…mutations of Smt3 acceptor sites… do not lead…"

REQUEST: L. 600: similar to L49, is Urm1 the modifier (the enzyme) or the modification

ANSWER: This sentence starts on line 589 in our document. We have changed "modifier" by "modification".

REQUEST: L. 772: The title of the review poses a direct question: chance or necessity? The authors could consider providing a more direct answer to that question in the last paragraph.

ANSWER: Very good point. In our opinion the question "chance or necessity?" is still unsolved. We are experimentally approaching the question using the Ub-eS31 and Ub-eL40 fusion proteins from yeast as working models, but so far we do not feel confident enough to favor one term (chance or necessity) over the other or even consider both as the most appropriate answer. As we emphasize in the last paragraph of the review only "future studies will undoubtedly provide valuable insights into the still unknown aspects […] of these ubiquitous proteins …".

REQUEST: Table S4: Abbreviations should be defined (in this table, ERAD appears to be the only one)

ANSWER: We have added at the end of the table "ERAD: ER-associated protein degradation".